# Associations between Dark Triad and Ambivalent Sexism: Sex Differences among Adolescents

**DOI:** 10.3390/ijerph17217754

**Published:** 2020-10-23

**Authors:** María Patricia Navas, Lorena Maneiro, Olalla Cutrín, Jose Antonio Gómez-Fraguela, Jorge Sobral

**Affiliations:** 1Department of Clinical Psychology and Psychobiology, Universidade de Santiago de Compostela, 15782 Santiago de Compostela, Spain; lorena.maneiro@usc.es (L.M.); olalla.cutrin@usc.es (O.C.); xa.gomez.fraguela@usc.es (J.A.G.-F.); 2Department of Politic Science and Sociology, Universidade de Santiago de Compostela, 15782 Santiago de Compostela, Spain; jorge.sobral@usc.es; 3Institute of Education and Child Studies, Leiden University, 2333 AK Leiden, The Netherlands; 4Global Center for Applied Health Research, Arizona State University, Phoenix, AZ 85004, USA

**Keywords:** Dark Triad, Machiavellianism, psychopathy, narcissism, Ambivalent sexism, adolescent

## Abstract

The Dark Triad traits (DT; Machiavellianism, psychopathy, and narcissism) have been repeatedly labeled as a constellation of traits that are characterized by a dishonest and self-focused approach to interpersonal relations. Personality psychologists suggest that these traits make some people more susceptible than others to intergroup bias, threat, and aggression. Thus, in order to delve into a psychological profile prone to accepting and justifying sexist attitudes, the aims of the current study were to analyze the presence of DT and sexist attitudes in a sample of 367 adolescents (*M_age_* = 15.12, *SD* = 0.88; 50.1% males), find out the relationships that DT has with both hostile and benevolent sexism, and analyze the relevant differences between sexes in these relationships. The results indicated higher scores in DT and Ambivalent sexism for males. The correlations of Machiavellianism with psychopathy, and psychopathy with narcissism revealed significantly higher associations in males than females. The structural equation modeling of the bifactorial model, characterized by a global latent factor that encompasses the common characteristics of DT—along with the three specific factors of Machiavellianism, psychopathy, and narcissism—showed that the global latent factor of DT was related to both hostile and benevolent sexism in males and females. Singularly, narcissism was related to benevolent sexism in males, and psychopathy was related to hostile sexism in females. Finally, this research discusses the implications of these results on the implementation of positive models of interpersonal relationships in adolescence towards dating violence prevention.

## 1. Introduction

As a result of social and political advances, the primary prevention of sexist attitudes in adolescence has become a priority of Spanish social policies. The National Institute of Statistics of Spain [1] states that 25% of teenage females feel monitored by their partner. One in three teenage males considers “controlling the couple’s schedules”, “preventing their partner from seeing their friends”, or “telling them what they can or cannot do” to be acceptable in some circumstances [2]. A sexist attitude specifically characterizes these behaviors, legitimized by differences in status and power between females and males [3].

### 1.1. Ambivalent Sexism

Sexism is understood as an attitude that implies a cognitive, affective, and behavioral response to a person due to his or her biological sex [4]. Allport provided the first attempt to explain sexism [5], defining sexism as a prejudice in which females are considered different and generally inferior. Therefore, according to this, females should adhere to gender-specific roles and social norms and behavior. This overt, discriminatory, and hostile sexism has survived and evolved over time into other forms of sexism, which are subtler, imperceptible, and more difficult to eradicate. The Ambivalent Sexism Theory of Glick and Fiske offers a validated theory about old and new forms of sexism [6]. According to this theory, sexism has a hostile and a benevolent component. Hostile sexism would largely coincide with the sexism that was described by Allport, since it is presented as an attitude with a negative sentiment based on heterosexual hostility. Benevolent sexism sustains an idealization of traditional gender roles; that is, females are “naturally” kind and more emotional, while males are “naturally” more rational and “tougher”, mentally and physically. Therefore, benevolent sexism is defined as a set of attitudes towards females with an equally sexist content, although their positive affectivity masks their true sexist sense [7]. Empirical evidence highlights sexist attitudes as a dynamic risk factor that explains the main element of dating violence in adolescence: the control or dominance of one partner over the other [8]. However, when considering the differences between sexes in college students, the literature shows unclear results: some studies reported more hostile and benevolent sexism in males than in females [9], while others did not find differences between sexes in terms of benevolent sexism [10].

### 1.2. Dark Triad Traits

To delve into positive role models, it is necessary to build a strong empirical knowledge base that demonstrates those variables that facilitate intergroup negativity and hinder the promotion of equality between males and females. Personality psychologists suggest that some qualities or traits make some people more susceptible than others to intergroup bias, threat and aggression [11]; and consequently, in recent years there has been a revival of interest in personality traits which predict prejudicial attitudes [12]. Into this framework, the literature has extensively used, with remarkable success, the Dark Triad of personality (henceforth DT) in order to designate a specific personality configuration characterized by manipulation and cynicism (i.e., Machiavellianism), callous social attitudes and impulsiveness (i.e., psychopathy), and vanity and self-centeredness (i.e., narcissism) [13]. Paulhus and Williams [14] proposed the label of DT while assuming the existence of a certain level of commonality between these three personality characteristics. That is, to define a personality profile that is characterized by manipulation, selfishness, and emotional coldness, especially focused on behaviors of domination, coercive control, and power [15]. An adaptive paradigm suggests that, despite its socially undesirable nature, this personality profile may have positive consequences for social, romantic, and vocational success [16] through manipulative social style or charm when dating people [17]. For this reason, although high levels of these traits exist in some clinical samples, they have also been found at the subclinical level. Moreover, regarding the differences between sexes, Muris et al. [18] conducted a meta-analysis, in which males scored higher on DT measures than females. This means that the reinforcement of DT behaviors in society benefits males over females, because they are consistent with traditional conceptualizations of male gender roles. Thus, from this adaptive perspective, darker personality variables may prove to be relevant in understanding sexism [19].

### 1.3. DT and Ambivalent Sexism

Few studies have analyzed the relationship between DT and Ambivalent sexism in adolescents. However, it is reasonable to assume that such a personality profile has an influence as a distal predictor of sexist attitudes and behaviors [20]. Previous research indicated that DT is related to attitudes toward rape in both sexes [21], facilitates an exploitive mating strategy [22]; and, in specific subclinical samples of adolescents, high levels of DT were associated with high levels of prejudice [23]. Likewise, studies document that sexism is a set of learned behaviors, and knowing the role that society plays in forming individuals with these undesirable personality characteristics can be useful in order to prevent or reduce it [19]. Although Machiavellianism, psychopathy, and narcissism might represent a single personality profile that encompasses the common characteristics [15], some authors consider them to be single constructs [24]. For example, previous studies indicate that narcissism has been associated with the “acceptance of rape “myth in males [25], and in females with tendencies to belong to norm [26]. Machiavellianism has been associated with promiscuity, hostile sexual attitudes, and various selfish and deceptive sexual tactics [27]; and, psychopathy has been associated with sexual coercion [28] and predatory sexual behaviors [29] in both sexes.

The DT is a cluster of subclinical traits that encompasses the reckless psychopath, the grandiose narcissist, and the strategic Machiavellian [28]. Although the sexist attitudes could be related with the common core of callousness and deception associated with all three DT traits, when a component of sexism (e.g., hostile) is associated with impulsive malevolence, psychopathy could be a complementary predictor. By contrast, when a component of sexism (e.g., benevolent) is associated with grandiosity and gallant behavior, narcissism could be implicated as a predictor and, when an outcome is associated with long-term strategic malevolence, Machiavellianism could be an associate predictor [30]. To summarize, all three traits can be used to hurt others for personal gain, but employing differential identity focus or strategies behaviorally differentiates the traits, so considering a bifactorial model of DT is coherent. These assertions have been substantiated using laboratory, survey, observational, meta-analytic, and behavioral genetic approaches in previous research [15]. As mentioned, the core of DT consists of high levels of deception and callousness [31]. However, each trait has unique characteristics beyond this core, which may be relevant to delve into what personality characteristics make some people more prone to hostile instead of benevolent attitudes. Apart from that, few studies have used a bifactorial model to research the characteristics of each trait beyond this core of DT that is related to sexism, and the differences between sexes in this context.

### 1.4. The Current Study

Therefore, based on the previous empirical evidence, it is a pending issue to know how Machiavellianism, psychopathy, and narcissism are related to Ambivalent sexism when considering differences between sexes. Consequentially, it appears that relating a bifactorial model of DT (characterized by a global latent factor that encompasses the DT’s common characteristics, along with the three specific factors of Machiavellianism, psychopathy, and narcissism) with both hostile and benevolent sexism in the adolescent sample is a productive approach [32]. Most research that has studied prejudice-related DT, has used clinical and subclinical samples of adults. Only 10.8% of published research on DT have used adolescent samples for their studies [24]. However, sexist patterns of interaction that facilitate tolerance towards violent behavior begin at an early age, when the first relationships in adolescence start to emerge. For this reason, the aim of this research has been to analyze the relationships between DT and Ambivalent sexism on a sample of adolescents when considering differences between sexes. Therefore, the current research entails the following components: (a) to analyze the presence of DT and sexist attitudes in an adolescent sample, to see if, as expected, males scores higher than females in both DT and Ambivalent sexism [9,10,23]; (b) to find out the relationships that DT has with both hostile and benevolent sexism, to see if, as expected, the global factor shows higher associations with sexist attitudes than with each component factor [15,16,31]; and finally, (c) to analyze the relevant differences between sexes in these relationships, in order to see if, as expected, males and females show a relationship between the global DT and both hostile and benevolent sexism [27], but only males show a relationship between narcissism and benevolent sexism.

## 2. Materials and Methods

### 2.1. Participants and Procedure

A statistical test of ANOVA was specified in G*Power 3 (Heinrich Heine Universität Düsseldorf, Düsseldorf, Germany) using a priory analysis and determining an anticipated medium effect size *f^2^* = 0.20 in order to calculate sample size. The program estimated a sample size of 328 participants, then the initial sample was composed of 421 participants. However, it was reduced after removing all of the participants with missing data in all study variables. The final sample was composed of 367 adolescents from six public high schools that were located in Galicia and Castilla—La Mancha (Spain), 50.1% males (n = 184) aged 13 to 18 years (*M_age_* = 15.15; *SD* = 0.63), and 49.9% females (n = 183) aged 14 to 18 years (*M_age_* = 15.10; *SD* = 0.68). The participation rates of the sample were high, as only 38 subjects decided not to participate in the study and, of the nine high schools that were contacted to become part of the research, six finally decided to participate in the study. For this reason, convenience sampling was performed in order to select state high schools from both autonomous communities, but the course in which adolescents were enrolled was randomly assigned to obtain data. 25.4% were enrolled in the third year and 43.9% in the fourth year of Compulsory Secondary Education, then 24.3% were enrolled in the first year and 6.4% in the second year of last grade of non-mandatory high-level education. The cultural and social characteristics of the sample are similar, because most youngsters come from middle and low-middle socioeconomic backgrounds.

The management team of each high school was in charge of requesting parental consent in order to carry out the assessment as an activity included in the study program of the 2018–2019 academic year, so the adolescents were invited to participate in the study by the high school management team. Only participants who had delivered parental consent to the management team, were older than 12 years, and displayed basic reading and writing skills were considered for their inclusion in the study. Afterwards, youth who consented to participating signed an informed consent form from the management center of each high school to assure their voluntary participation without any type of compensation. Confidentiality and anonymity were ensured in accordance with the 1964 Helsinki declaration [33] and the guidelines of University ’s Bioethics Committee. Finally, the participants completed a forty-five-minute questionnaire in group sessions as part of the course in the presence of a member of the research team, which provided the students with the opportunity to ask questions or receive support if necessary.

### 2.2. Measurements

Dark Triad traits: the Spanish version of Dirty Dozen scale (DD) [34], translated and validated into Spanish by Maneiro et al. [31], was been used to assess DT. This inventory measures each component of DT with only four items each, giving rise to a total of 12 items using response options on a five-point Likert-type scale ranging from 1 (totally disagree) to 5 (totally agree). The scale provides a global score for DT as well as a score for each component of the triad: Machiavellianism (e.g., “I tend to manipulate others to get my way”, *α* = 0.80), psychopathy (e.g., “I tend to lack remorse”; *α* = 0.70), and narcissism (e.g., “I tend to want others to admire me”; *α* = 0.84). This instrument showed an acceptable internal consistency, with Cronbach’s alpha for a global score of 0.85. The confirmatory factor analysis performed to know the fit of the bifactorial model to the structure of Dirty Dozen scale in Spanish language showed good parameters (*χ*^2^ = 77.29, *df* = 42, *p* < 0.001; SRMR = 0.03; RMSEA = 0.04; CFI = 0.98). After that, configural, metric, and scalar invariances were tested across the groups [34,35]. The results proved that configural (*χ*^2^ = 98.147, *df* = 84, *p* > 0.05; RMSEA = 0.03; SRMR = 0.03; CFI = 0.98) and metric (*χ*^2^ = 157.292, *df* = 112, *p* < 0.01; RMSEA = 0.05; SRMR = 0.08; CFI = 0.96) were invariant across sexes (ΔCFIs < 0.01), but not scalar (*χ*^2^ = 145.595, *df* = 104, *p* < 0.01; RMSEA = 0.05; SRMR = 0.07; CFI = 0.96).

Ambivalent sexism: Ambivalent Sexism Inventory for Adolescents (ISA scale) [35]; contains 20 items that were divided into two subscales that measure hostile sexism and benevolent sexism, using a response format from 0 (totally disagree) to 5 (totally agree); higher scores reflect more sexist attitudes. The first 10 items measure hostile sexism (e.g., ‘‘Young men should exert control over who their girlfriends interact with’’), and the remaining 10 items measure benevolent sexism (e.g., ‘‘A boy will feel incomplete if he is not dating a girl’’). Internal consistency, as obtained in our study, was 0.86 for the hostile sexism subscale and 0.82 for the benevolent sexism subscale.

### 2.3. Data Analysis

Data analyses were conducted on IBM SPSS Statistics 23 (IBM, New York, NY, USA), and Mplus v.7 (Muthén & Muthén, Los Angeles, CA, USA) was used for the analyses of the structural equation modeling (SEM). Firstly, one-way ANOVAs were performed in order to analyze differences between sexes in all of the study variables. Subsequently, partial eta squared was used to analyze the magnitude of sex differences following the benchmarks to define small (η_p_^2^ = 0.01), medium (η_p_^2^ = 0.06), and large (η_p_^2^ = 0.14) effect sizes. Secondly, zero-order correlations were used to assess the relationship between DT and both hostile and benevolent sexism, and Fisher’s Z transformations were performed in order to analyze the differences between sexes on the correlation measures. Finally, structural equation modeling was tested in order to know the contribution of the global DT´s latent factor and the three latent factors that are associated with the subscales of the Dirty Dozen on both hostile and benevolent sexism. Robust maximum likelihood estimation as well as the goodness of fit indexes *χ*^2/^*df*, CFI, SRMR and RMSEA were used for model estimation [36]. The following criterion were considered for an optimal fit *χ*^2^/*df* < 2–3, CFI > 0.95, RMSEA < 0.06, and SRMR < 0.05; and, for an acceptable or reasonable fit *χ*^2^/*df* < 4, CFI > 0.90, RMSEA < 0.08, and SRMR < 0.08.

## 3. Results

### 3.1. Descriptive Statistics and Correlations

Table 1 displays descriptive statistics, including mean and standard deviations, the internal consistency of all study variables for each sex, as well as the differences between sexes in all of the study variables. The ANOVA results indicated that males scored higher than females in all of the variables, including DT and Ambivalent sexism. Partial eta squared showed small effects of sex influence on DT traits and large effects on hostile sexism where specifically males show the highest scores.

Table 2 presents the bivariate correlation coefficients and Fisher’s Z transformation performed to analyze differences between sexes. The results showed no significant associations between age and any study variable. The associations between the majority study variables were direct and significant in adolescents. For females, the association of psychopathy with narcissism was not significant. For males, the associations of Machiavellianism and psychopathy with benevolent sexism were not significant. The coefficients of psychopathy with narcissism revealed significantly higher associations coefficients in males than in females (*p* < 0.05). The association coefficients between Machiavellianism and psychopathy also presented significantly higher associations in males than females (*p* < 0.001).

### 3.2. SEM Between DT and Ambivalent Sexism and Differences between Sexes

After these analyses, SEM was used in order to evaluate the existence of a significant relationship between the bifactorial model (that includes a latent factor of global DT and three latent factors associated with each DT traits) and hostile and benevolent sexism. The models obtained acceptable parameters (*χ*^2^/*df* < 4, RMSEA and SRMR between 0.08 and 0.10) [37], for males (*χ*^2^ = 836.69, *df* = 946, *p* < 0.001; SRMR = 0.09; RMSEA = 0.06) and females (*χ*^2^ = 870.72, *df* = 946, *p* < 0.001; SRMR = 0.09; RMSEA = 0.06). The results that are displayed in Figure 1 showed that in males, the global DT was related with hostile and benevolent sexism while narcissism was related to benevolent sexism. The results that are presented in Figure 2 showed that, in females, the global DT was related with hostile and benevolent sexism, while psychopathy was related to hostile sexism.

## 4. Discussion

There has been discussion regarding whether personality characteristics make some people more prone to sexist attitudes than others. Some research has not supported this idea, but this has not prevented others from compiling a long list of variables of individual differences that are related to sexist attitudes [38].

### 4.1. Presence of the DT Traits and Ambivalent Sexism in Adolescents

The results demonstrate that adolescents show higher scores in benevolent sexism than hostile sexism. This type of sexism, characterized by a positive affectivity, has been associated with myths of romantic love, such as considering jealousy as a sign of love and care [39]. These attitudes can lead adolescents to overlook violent behavior (e.g., control the couple’s schedules or requiring constant disclosure) in their first romantic relationships, increasing the likelihood of dating violence [8]. Therefore, our findings are consistent with those previously found in Diaz-Aguado et al. [40] that warn of the need for preventive intervention that is focused on the detection and analysis of microsexism, paternalistic and protective behaviors, as well as the uncritical acceptance of traditional gender roles that are masked with positive affectivity.

When considering the differences between sexes, the results are consistent with previous research [9], in which males show higher scores in both hostile and benevolent sexism than females. Similarly, DT traits have shown clear differences between sexes, replicating previous findings, where males are more prone to antisocial personality traits than females [41]. The associations of Machiavellianism with psychopathy and narcissism with psychopathy scored significatively higher in males than females. These results are consistent with previous research, suggesting this is a stable effect [42]. Likewise, consistent with the developmental stage of the sample, higher average scores in narcissism have been found than Machiavellianism and psychopathy. The achievement of self-identity is a fundamental task in adolescence and, for this, young people are interested in standing out visibly, being recognized and, to some extent, being admired [43]. Therefore, to achieve this, possibly a usual strategy is the acceptance of traditional gender roles, since the results highlight relationships that narcissism has with both hostile and benevolent sexism.

### 4.2. Relationships between DT with Hostile and Benevolent Sexism

Referring to the association between the bifactorial model of DT and Ambivalent sexism, the results show better associations between global DT and both hostile and benevolent sexism than each of three constitutive factors (i.e., Machiavellianism, psychopathy, and narcissism). That is to say that individuals with high scores in the global DT also obtain high scores in hostile and benevolent sexism. These results are fully consistent with previous literature. Hostile and benevolent sexism act as an articulated system of rewards and punishments that determines a female’s position in society. These findings can be explained with the evolutionary-adaptative paradigm that is mentioned above [15]. Like sexist attitudes, DT is a changing condition, a product of society that reinforces and facilitates seductive behaviors and interpersonal manipulation styles. At least in part, DT represents a general and stable way of going through life, and sexist attitudes could be a specific domain where these basic personality trends in DT are manifested [19]. The high associations between DT and Ambivalent sexism justify and validate this proposal, where sexism is reformulated and legitimized as this group of personality traits. This adaptive understanding leads us to prevent or reduce behaviors that are characteristic of DT, recognizing these attributes as chosen behaviors instead of personality traits that limit us to progressing towards a more equal and less sexist society.

Nonetheless, this association has occurred in different ways for each sex. In males, high scores on the global DT are associated with high scores of hostile sexism, while high scores on the global DT, as well as high scores on narcissism, are associated with high scores of benevolent sexism. This narcissistic feature that provides success and (inter) personal satisfaction through the adoption of gallant and dominating behaviors, can operate by diminishing or masking the most socially undesirable behavioral aspects [44]. They also play a decisive role in the genesis, reproduction, and maintenance of that positive affective tone that masks and reinforces a system of equally sexist attitudes through a hegemonic masculinity. Furthermore, as we expected, the global DT has shown a high association with hostile sexism: insensitivity, selfishness, egocentrism, and malevolence in their interpersonal relationships, as well as the probable intention of punishing females who reject prevailing cultural norms of expected femininity. Thus, hostile and benevolent attitudes are intimately related, and they may simultaneously appear in the same subject, constituting a complex socio-cognitive mechanism of social control [45].

Females, like males, can also have sexist attitudes [6]. The cultural transmission of sexism to females may reflect their tendency to embrace or reject prevailing cultural norms and traditional gender roles. High scores on the global DT in females are associated with high scores of benevolent sexism. This result is consistent with previous research, in which individuals who exhibited malevolent personality traits were also more likely to subscribe to traditional gender norms and engage in more selfish behaviors that increase the probability of interpersonal success, especially vocationally and romantically [13,15,19]. High scores on the global DT, as well as high scores on psychopathy, are associated with high scores of hostile sexism. These results support the idea that, in certain contexts, females show some willingness to act unkindly and irresponsibly towards other females who decide to challenge traditional gender roles [15] or try to alter the power relations between males and females. In this way, implicit attitudes, such as justifying rape or not blaming the aggressor, would be more frequent if these individuals showed less empathetic predisposition [20].

In order to implement positive models to reduce dating violence in interpersonal adolescent relationships, it is advisable to work during childhood and adolescence on the set of prototypical DT behaviors that are involved in sexist attitudes and emerge in early relationships [19]. Namely, actions in the domains of changing behavioral habits and routines (e.g., programming reinforcements for empathic behaviors), cognitive restructuring of biases and beliefs (e.g., critical analysis of toxic beliefs), and emotional regulation (e.g., regulation of anger associated with partner situations). The in-depth implementation of these intervention strategies is the only hope to palliate the influence of the DT on sexism and dating violence. Furthermore, the results show, due to its balance between economics and psychometric indicators [32], the usefulness of adding the DD instrument for improving the predictive assessment of dating violence risk, by detecting early high levels of DT.

### 4.3. Limitations and Future Research

The current research contributes to developing a psychological profile prone to accepting and justifying sexist attitudes, at an early age, and in community samples. Most previous investigations have exclusively examined Ambivalent sexism from a social-psychology perspective. Our work considers the recommendations of Akrami et al. [12], where combining the influence of both personality- and social-psychology is relevant in explaining sexist attitudes. However, this research is not free of limitations: the current study is limited by cross-sectional data. This mainstreaming does not allow for us to propose precise relations of causality between our variables. Furthermore, non-probabilistic sampling methods were used for participant selection, which limits the representativeness of the sample and, consequently, the generalization of the results to the entire population. Future longitudinal studies should clarify temporal sequences of determination of some variables on others throughout the socialization process. Likewise, future research might analyze the directly observed relationship of DT with explicit behaviors of dating violence.

## 5. Conclusions

In conclusion, the current study has fulfilled the objective of highlighting the strong association that sexist attitudes have with a personality profile that encompasses the common characteristics of the three specific factors of Machiavellianism, psychopathy, and narcissism. Deepening the knowledge of the personality traits could help to implement positive models in a more effective way considering the individual differences of adolescents. To know which attitudes prevail in them has also allowed for us to know what type of strategies are appropriate for the education and prevention of sexist behaviors in both sexes of the sample. Additionally, it is of special interest to have shown it in a normalized community sample of adolescents. It is important to fight during childhood and adolescence against a set of toxic cognitions that are involved in sexist attitudes and the DT construct. Furthermore, DT is a personality configuration that must be noted and analyzed by normal people in everyday contexts.

## Figures and Tables

**Figure 1 ijerph-17-07754-f001:**
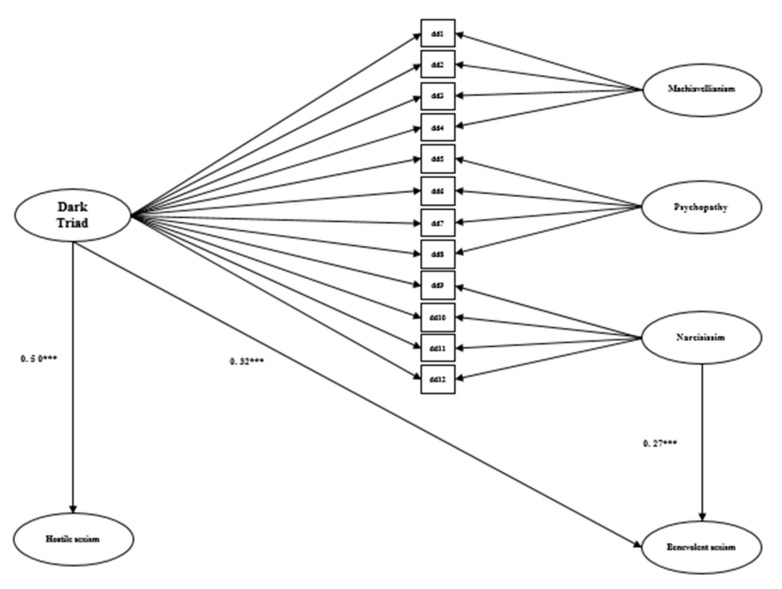
Standardized coefficients of regression statistically significant at *p* < 0.05 between Dark Triad and Ambivalent sexism in males. *** *p* < 0.001.

**Figure 2 ijerph-17-07754-f002:**
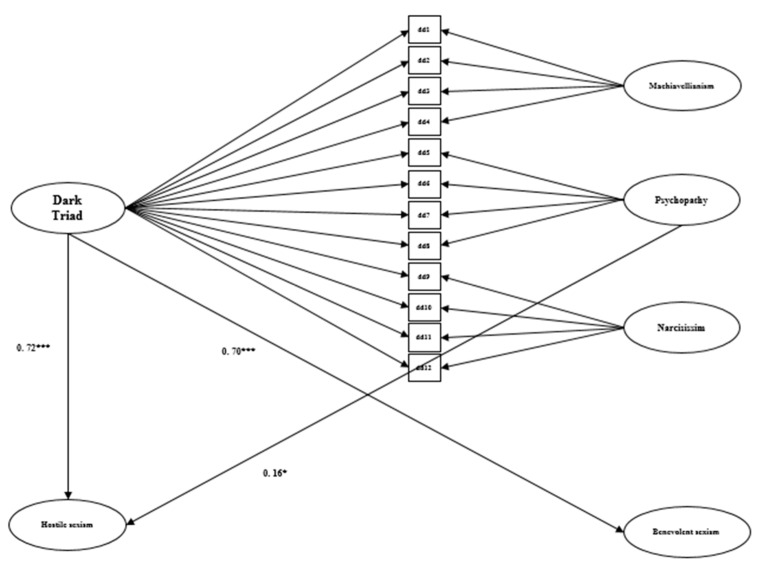
Standardized coefficients of regression statistically significant at *p* < 0.05 between Dark Triad and Ambivalent sexism in females. * *p* < 0.05; *** *p* < 0.001.

**Table 1 ijerph-17-07754-t001:** Descriptive statistics and differences among males and females in Dark Triad (DT) traits and Ambivalent sexism.

	Range	Males		Females		*F*	η_p_^2^
		*M* (*SD*)	*α*	*M* (*SD*)	α		
Machiavellianism	4–18	9.17 (3.73)	0.81	8.35 (3.73)	0.76	4.48 *	0.01
Psychopathy	4–15	7.34 (3.16)	0.71	6.25 (2.49)	0.67	13.46 **	0.04
Narcissism	4–19	10.17 (4.08)	0.85	8.49 (3.78)	0.80	16.74 **	0.05
Hostile sexism	10–37	26.81 (10.08)	0.85	18.02 (7.33)	0.80	91.18 ***	0.20
Benevolent sexism	10–54	28.29 (9.52)	0.79	22.92 (9.40)	0.83	29.53 ***	0.07

*Note*. * *p* < 0.05, ** *p* < 0.01, *** *p* < 0.001. η_p_^2^ = partial eta squared effect size.

**Table 2 ijerph-17-07754-t002:** Bivariate correlations between all of the study variables according to sex.

	1	2	3	4	5	6
Age	-	−0.04	−0.12	−0.01	0.01	0.08
2. Hostile sexism	−0.04	-	0.52 **	0.32 **	0.22 **	0.39 **
3. Benevolent sexism	−0.06	0.57 **	-	0.12	0.12	0.35 **
4. Machiavellianism	−0.01	0.41 **	0.34 **	-	**(0.55 **)**	0.54 **
5. Psychopathy	0.01	0.38 **	0.29 **	**0.26 ****	-	**(0.33 **)**
6. Narcissism	0.02	0.39 **	0.35 **	0.52 **	**0.11**	**-**

*Note.* The coefficients above the diagonal correspond to males, and scores below the diagonal correspond to females. ** *p* < 0.01. The coefficients in bold present significant differences between males and females (*p* < 0.05).

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
