# Peer review of "Associations between Dark Triad and Ambivalent Sexism: Sex Differences among Adolescents"

_ijerph, 2020, doi:10.3390/ijerph17217754_

Round 1

Reviewer 1 Report

Dear authors

This research "Associations between dark triad and ambivalent sexism: sexual differences among adolescents" examines the presence of dark triad and sexist attitudes in a sample of adolescents, discovers the relationships that have dark triad with hostile and benevolent sexism, and analyzes sexual differences.

I believe that the study contributes significantly to this area of ​​research and that it also has important practical implications for the prevention of sexist attitudes and dating violence in adolescents.

However, I have several aspects that I would like the authors to think about and include if possible:

Abstract: I suggest including the 3 objectives of the study: to examine the presence of DT and ambivalent sexism, the relationships between DT and ambivalent sexism and sex differences. This would better understand all the results presented in the abstract.

Keywords: If possible, include one more word: “adolescent”.

Introduction:

  • With respect to subsection 1.1. “Ambivalent sexism” the authors point out “However, considering the sex differences, the literature shows unclear results: some studies reported more Hostile and Benevolent Sexism in men than in women [9]; while others found no sex differences in relation to Benevolent Sexism [10]”. Could you add more information about these two studies, for example, the type of sample?
  • With respect to subsection 1.2. “Dark Triad traits”, It would be necessary to mention studies with subclinical samples that analyze DT and found gender differences. This aspect is important because it is part of the hypotheses raised.
  • With respect to subsection 1.3. "DT and ambivalent sexism" would facilitate reading if DT always preceded Ambivalent sexism in the writing, i.e."Few studies have analyzed the relationship between DT and ambivalent sexism in adolescents". I suggest the same in the presentation of results and discussion.
  • The following data would be better located in subsection 1.2. “Furthermore, with regard to sex differences, Muris et al. [23] performed a meta-analysis in which men scored higher on TD measures than women”.
  • The 3 objectives are correctly. However, only in the first objective are the expected results indicated. It would be necessary to indicate the expected results in objectives 2 and 3. It would also be convenient to include the citations of studies that support these hypotheses.

Participants and procedure:

  • It is mentioned how the samples were recruited. How high were the participation rates?
  • It has not been mentioned how parental consent was obtained for those under 18 years of age.
  • It would be necessary to include the approximate duration of the group evaluation sessions and the dates on which this evaluation was carried out.
  • It is convenient to indicate how the sample size was calculated for this study.
  • Inclusion and exclusion criteria should be added.
  • Measures: It would be necessary to include information on the meaning of the scores.

Results:

  • In line with what was suggested above, I recommend that the same order be followed in the presentation of results, first DT and then Ambivalent sexism, also in table 1.
  • I am not able to understand well the presentation of results referring to the relationships between the different components of DT and the gender differences in these relationships. If these results are included, the importance and contribution they make should be justified.
  • In the note of figures 1 and 2 there is an unnecessary acronym DT.

Discussion:

The discussion begins with data that justifies the study. I believe that this information should be at the end of the introduction, just before the objectives are stated.

Subsection 4.1.

  • The authors state: “The results support that adolescents show higher scores in Benevolent sexism than Hostile sexism”. Although it is not the objective of this study, the authors provide this information. In the case of maintaining it, it is convenient to contrast these data with other studies in which it can be assumed that they find the same results.
  • It indicates “Considering sex differences, males show higher scores in both Hostile and Benevolent sexism than females”. This data is not accompanied by contrast with other studies.

Subsection 4.2.

  • In line with what has been indicated on previous occasions, I suggest that the order in the title and in the text be: "Relationships between DT with hostile and benevolent sexism"
  • The association between DT and sexism is mentioned very frequently but it would help to better understand the results if the meaning of the associations is written in this positive case, for example, individuals with high scores in DT also obtain high scores in hostile sexism and benevolent.

The authors have pointed out the practical implications of the results. I suggest that these practical implications be explored further. In addition, throughout the study some of these implications have been identified, including the abstract that indicates implications for the prevention of violence in dating relationships, and other sections indicate implications for the prevention of sexist attitudes in adolescents. It is convenient that the discussion includes the implications indicated in other sections.

Finally, I wanted to point out undifferentiated use of capital letters in some concepts: i.e., Machiavellianism, psychopathy…

I hope these comments can be of help to the authors.

Reviewer 2 Report

I have reviewed the manuscript entitled “Associations between Dark Triad and Ambivalent 2 sexism: sex differences among adolescents”. The main objective of this study was to test the relationships of Dark Triad with Ambivalent sexism, as well as analyze differences in this relationship between sexes on a sample of adolescents. Generally, the manuscript makes a very valuable contribution to the field, including current and relevant references. The research design is appropriate. The methods are clearly described. However, the manuscript must be revised based on English language grammar. There are words translated into English that are not often used in the scientific literature. Thus, moderate English changes are required.

The introduction provides sufficient background.

Material and Methods.

-Participants and procedure section could be improved.

-Line 131: add cite (and reference)

Results

Table 2. The coefficients in brackets are confusing. Please add values based on males scores above the diagonal, and values based on females scores below the diagonal.

Discussion and conclusions are supported by results.

Reviewer 3 Report

In the manuscript “Associations between Dark Triad and Ambivalent sexism: sex differences among adolescents” the authors analyze the relationships between the dark triad and hostile and benevolent sexism in a sample of 367 Spanish adolescents. The manuscript covers important topics. However, some changes must be made before publication, as there are some shortcomings that detract from the quality of the manuscript.

Abstract:

On page 1, lines 24-25 reads, “The correlations of Machiavellianism with psychopathy, and psychopathy with narcissism revealed significant differences between males and females.” The significant differences were between males and females should be specified.

On line 28 it says "laten", it should say "Latent"

Introduction:

The Introduction section should be revised and explain in more depth the theoretical basis of the work so that it is clear to the reader a firm sense of what was done and why.

Page 3, lines 102-103 says: “Given the inconsistencies in previous research, it is a pending issue to know how each personality construct relate Ambivalent sexism considering sex differences.” It should be specified which personality constructs they refer to.

Measurements:

The description of the Ambivalent sexism inventory should be reviewed and clearly specified how each of the dimensions were assessed.

Page 4, lines 148 to 154 says: “This scale is composed of 20 items, which provide information about three differentiated dimensions: Hostile sexism (e.g., “Males must control who they meet they relate with their girlfriends”), Benevolent sexism (e.g., “Females must be loved and protected for the males”) and Ambivalent sexism (global score composed by both Hostile and Benevolent sexism) through a 6-point Likert-type scale ranging from 0 (totally disagree) to 5 (totally agree). For the current study, only the subscales of Hostile and Benevolent sexism were included in the analysis.” It is not understood why the dimension of Ambivalent sexism was not included, since the title of the manuscript is “Associations between Dark Triad and Ambivalent sexism: sex differences among adolescents”

References:

The references section should be revised as some appear incomplete. For example, on Page 8, lines 315-316 puts “INE. Estadística de Violencia Doméstica y Violencia de Género [Statistics on Domestic and Gender Violence]. Available online:                                       ”

Reviewer 4 Report

I found this article to be too confusing to sufficiently understand whether it is significant.  It was not clearly written and I believe it should have been checked for grammar and meaning before its submission.  There are too many parts of the paper where I had to take the time to try to decipher the meaning.  

I'm not sure whether it was the flawed writing or the contents themselves, but I was unable to see the import of the work.  

In other words, it was not clear to me how the research can be used other than describing issues surrounding dating violence in the terms of the so-called Dark Triad traits.  Thus, I remain unsure of how framing issues surrounding the description and remediation of dating violence are enhanced by the authors' work.   

I do not have the time to go through and edit the paper, but I will provide a sample of some of the problems with the writing:

L 16: "has" should be "have"

L 28 "laten"   should this be latent?

L 95 profile that encompasses the common characteristics [15]; some authors . . .  [semicolon should be a comma]

L 102-3: Given the inconsistencies in previous research, it is a pending issue to know how each personality construct relate Ambivalent sexism considering sex differences.   [I don’t understand and have no idea how to fix this]

L 107:  To be grammatically correct, change this: “Therefore, the current research has the purpose of":  to "Therefore, the current research entails the following components:"

L 126-8 The performance of this activity did not have any type of compensation and all procedures approved by the Bioethics Committee of the University were assured throughout this research.

change to:

The performance of this activity did not have any type of compensation and all procedures were approved by the Bioethics Committee of the University.

L 128-131

After having signed an informed consent from management center of each high schools, the participation was voluntary and only individuals who consented to take part in the study were included as participants. Confidentiality and anonymity were ensured following the1964 Helsinki declaration.

change to:

Youth who consented to participate signed an informed consent form from the management center of each high school to assure their voluntary participation. Confidentiality and anonymity were ensured in accordance with the 1964 Helsinki declaration.

L 178-9 SHOULD BE: The results showed no significant associations between age and any study variable.

L 206 There has been discussed whether . . .

SHOULD BE: There has been discussion of whether . . .

Line 222: micromachisms ???

Line 250  way should be WAYS

Line 263. Females, like men,

Say either: Women, like men,

OR Females, like males,

L 279 combine SHOULD BE COMBINED OR COMBINING

L 280: However, this research has not been exempted of limitations

SHOULD BE: However, this research is not free of limitations

L 282: "venture" should be “propose”

L 286.  Add “the” directly observed . . .

L 297-8. against set of toxic cognitions involved in sexist attitudes and nature of DT construct.

CHANGE TO:

against a set of toxic cognitions involved in sexist attitudes and the DT construct.

In addition, I would also add to the Abstract on lines 30-1:

Finally, the implications [DESCRIBE the implications briefly here] of these results for dating violence prevention in adolescents are discussed.

Round 2

Reviewer 3 Report

The manuscript “Associations between Dark Triad and Ambivalent sexism: sex differences among adolescents” deals with a relevant topic from a theoretical and applied perspective. The revised manuscript has improved significantly from the initial version, so I think it can be published.

Reviewer 4 Report

The paper is greatly improved.  I appreciate all the work the authors did to make the paper more readable and to clarify its significance.

I only noticed one grammatical problem:

Lines 316-318: Namely, actions in the domains of changing behavioral habits and routines (e.g., programming reinforcements for empathic behaviors), cognitive restructuring of biases and beliefs (e.g., critical analysis of toxic beliefs), and emotional regulation (e.g., regulation of anger associated with partner situations).

The excerpt above is not a complete sentence; it's lacking a verb.